# Novel Genetically Encoded Bright Positive Calcium Indicator NCaMP7 Based on the mNeonGreen Fluorescent Protein

**DOI:** 10.3390/ijms21051644

**Published:** 2020-02-28

**Authors:** Oksana M. Subach, Vladimir P. Sotskov, Viktor V. Plusnin, Anna M. Gruzdeva, Natalia V. Barykina, Olga I. Ivashkina, Konstantin V. Anokhin, Alena Y. Nikolaeva, Dmitry A. Korzhenevskiy, Anna V. Vlaskina, Vladimir A. Lazarenko, Konstantin M. Boyko, Tatiana V. Rakitina, Anna M. Varizhuk, Galina E. Pozmogova, Oleg V. Podgorny, Kiryl D. Piatkevich, Edward S. Boyden, Fedor V. Subach

**Affiliations:** 1National Research Center “Kurchatov Institute”, Moscow 123182, Russia; subach_om@nrcki.ru (O.M.S.); witkax@mail.ru (V.V.P.); annadronova@mail.ru (A.M.G.); Ivashkina_OI@nrcki.ru (O.I.I.); nikolaeva_ay@nrcki.ru (A.Y.N.); igra-voina@yandex.ru (D.A.K.); annavlaskina@yandex.ru (A.V.V.); Lazarenko_VA@nrcki.ru (V.A.L.); taniarakitina@yahoo.com (T.V.R.); 2Institute for Advanced Brain Studies, M.V. Lomonosov Moscow State University, Moscow 119991, Russia; vsotskov@list.ru (V.P.S.); k.anokhin@gmail.com (K.V.A.); 3P.K. Anokhin Research Institute of Normal Physiology, Moscow 125315, Russia; n.barykina@nphys.ru; 4Bach Institute of Biochemistry, Research Center of Biotechnology of the Russian Academy of Sciences, Moscow 119071, Russia; boiko_konstantin@inbi.ras.ru; 5M.M. Shemyakin and Yu.A. Ovchinnikov Institute of Bioorganic Chemistry, RAS, Moscow 117997, Russia; olegpodgorny@inbox.ru; 6Research and Clinical Center of Physical-Chemical Medicine of Federal Medical Biological Agency, Moscow 119435, Russia; annavarizhuk@gmail.com (A.M.V.); pozmge@gmail.com (G.E.P.); 7Center for Precision Genome Editing and Genetic Technologies for Biomedicine, Moscow 119435, Russia; 8Center for Precision Genome Editing and Genetic Technologies for Biomedicine, Pirogov Russian National Research Medical University, Moscow 117997, Russia; 9N.K. Koltzov Institute of Developmental Biology, RAS, Moscow 119334, Russia; 10Massachusetts Institute of Technology, Cambridge, MA 02139, USA; kiryl.piatkevich@gmail.com (K.D.P.); esb@media.mit.edu (E.S.B.); 11School of Life Sciences, Westlake University, Hangzhou 310024, China

**Keywords:** genetically encoded calcium indicator (GECI), protein engineering, calcium imaging, crystal structure, NCaMP7, high brightness, fluorescent protein

## Abstract

Green fluorescent genetically encoded calcium indicators (GECIs) are the most popular tool for visualization of calcium dynamics in vivo. However, most of them are based on the EGFP protein and have similar molecular brightnesses. The NTnC indicator, which is composed of the mNeonGreen fluorescent protein with the insertion of troponin C, has higher brightness as compared to EGFP-based GECIs, but shows a limited inverted response with an ΔF/F of 1. By insertion of a calmodulin/M13-peptide pair into the mNeonGreen protein, we developed a green GECI called NCaMP7. In vitro, NCaMP7 showed positive response with an ΔF/F of 27 and high affinity (K_d_ of 125 nM) to calcium ions. NCaMP7 demonstrated a 1.7-fold higher brightness and similar calcium-association/dissociation dynamics compared to the standard GCaMP6s GECI in vitro. According to fluorescence recovery after photobleaching (FRAP) experiments, the NCaMP7 design partially prevented interactions of NCaMP7 with the intracellular environment. The NCaMP7 crystal structure was obtained at 1.75 Å resolution to uncover the molecular basis of its calcium ions sensitivity. The NCaMP7 indicator retained a high and fast response when expressed in cultured HeLa and neuronal cells. Finally, we successfully utilized the NCaMP7 indicator for in vivo visualization of grating-evoked and place-dependent neuronal activity in the visual cortex and the hippocampus of mice using a two-photon microscope and an NVista miniscope, respectively.

## 1. Introduction

Genetically encoded calcium indicators (GECIs) developed from the derivatives of the EGFP fluorescent protein are indispensable tools for the visualization of neuronal activity in vivo using both one- and two-photon microscopy [1]. Despite the great efforts in the development of green GECIs, their brightness is mainly limited by the brightness of the EGFP fluorescent protein (FP).

Recently, the brightest monomeric green fluorescent protein, mNeonGreen, has been developed and applied for the engineering of different types of indicators. mNeonGreen has been engineered from the protein derived from *Branchiostoma lanceolatum* and has a 3-fold higher molecular brightness in terms of quantum yield and extinction coefficient product as compared to the EGFP protein [2]. Due to its high brightness and monomeric state, mNeonGreen has been successfully applied to develop genetically encoded indicators for membrane potential [3], cAMP [4], caspase activity [5], and hydrogen peroxide [6]. However, only two calcium indicators, GeNL(Ca^2+^) [7] and NTnC [8], utilize mNeonGreen as a fluorescence moiety. GeNL(Ca^2+^) is a bioluminescent calcium indicator that has the main limitation of a slow rate of luminescence imaging and does not allow fast in vivo calcium imaging. The NTnC green calcium indicator utilizes mNeonGreen as a fluorescence moiety and troponin C as a calcium-binding moiety and has molecular brightness 1.6-fold higher than the brightness of EGFP [8]. However, NTnC exhibits an inverted response to calcium ions and has the modest dynamic range (ΔF/F of 1). A positive response would be advantageous for long-term and three-dimensional (3D) calcium imaging of large-scale neuronal populations in vivo [9].

Two design strategies are mainly used for the development of calcium indicators based on single FPs [10]. Most of the developed single FP-based calcium indicators, such as GCaMP [11] and pericam [12], are based on the concept of circular permutation, in which two portions of FP polypeptide are flipped around with new *N*- and *C*-termini, which are located in the β-strand 7 of FPs [10,13], and the calcium-binding part is fused to new *N*- and *C*-termini. The other strategy for developing single FP-based calcium indicators is the insertion of the calcium-binding domain into the β-barrel of FPs [10]. Several calcium indicators, including NTnC [8], iYTnC2 [14], YTnC [9], NIR-GECO1 [15], GAF-CaMP2 [16], and ncpGCaMP6s [17], were constructed by inserting the calcium-binding domain into FPs. Thus, the NTnC calcium indicator was constructed by inserting troponin C between positions 145 and 146 of mNeonGreen [8].

The NTnC-like design is the best choice for the development of novel bright calcium indicators for several reasons. The NTnC-like design with the insertion of the Ca^2+^-binding domain inside the mNeonGreen protein preserves both the *N*- and *C*-termini of the indicator on the fluorescent moiety side. In contrast, the GCaMP-like design suggests the presence of a short M13-like peptide and CaM calcium-sensing moieties on its *N-* and *C*-termini, respectively. In the latter case, the *N*- or *C*-terminal fusions to the calcium indicator can affect its dynamic range and affinity to calcium ions [18]. In previous work, we also demonstrated that the NTnC-like design was preferential over the GCaMP-like design in terms of localization, brightness, and calcium sensitivity in the *N*-terminal fusions with other proteins [9].

Recently, the NTnC-like design was successfully applied in the ncpGCaMP6s calcium indicator with the insertion of a calmodulin CaM/M13-like peptide pair as the Ca^2+^-binding domain into the EGFP fluorescent protein [17]. The ncpGCaMP6s indicator demonstrated a high dynamic range (ΔF/F of 25) and a positive response to calcium ions. This success prompted us to employ a similar CaM/M13-like peptide pair for the development of an mNeonGreen-based indicator with significantly better characteristics as compared with the NTnC indicator developed earlier. 

Herein, we report the engineering of a series of the brightest green GECIs developed by the insertion of the CaM/M13-peptide Ca^2+^-binding domain into the mNeonGreen fluorescent protein. This indicator was named NCaMP7 and was characterized by a 1.8-fold higher molecular brightness as compared to the brightness of the mEGFP protein, which makes NCaMP7 the brightest green calcium indicator among all GECIs reported to date. The NCaMP7 indicator demonstrated a positive fluorescence response to Ca^2+^ ions, with a high dynamic range (ΔF/F of 27) and high affinity to calcium ions of 125 nM in vitro. NCaMP7 had higher and similar pH stability in apo- and sat-states compared to the same states for the GCaMP6s GECI in vitro, respectively. According to the stopped-flow experiments in vitro, NCaMP7 demonstrated calcium-association/dissociation kinetics similar to that for GCaMP6s. We also developed and characterized variants of NCaMP7, called NCaMP4, NCaMP9, and NCaMP10, with increased and lowered affinity to calcium ions; however, they demonstrated lower ΔF/F response in vivo. Purified NCaMP7 was crystallized in a calcium-bound state; its structure was solved at 1.75 Å resolution and thoroughly analyzed. According to the fluorescence recovery after the photobleaching (FRAP) experiments, in a short time scale under low calcium concentrations, NCaMP7 demonstrated a 2.5-fold higher mobility than the GCaMP6s GECI. NCaMP7 demonstrated an ΔF/F of 10 in HeLa mammalian cells, similar to that for GCaMP6s. NCaMP7 robustly monitored spontaneous neuronal activity in primary cultured neurons. We successfully applied NCaMP7 for fast calcium imaging in the hippocampus of freely moving mice, with detection of non-specific or specific space-evoked place cells activity using a one-photon nVista miniscope. Finally, NCaMP7 allowed the robust detection of grating-evoked neuronal activity in the visual cortex of the mouse fixed under a two-photon microscope.

## 2. Results and Discussion

### 2.1. Development of Bright Green Fluorescent Positive Calcium Indicators Based on the mNeonGreen Protein 

To develop a bright green calcium indicator, we inserted a CaM/M13-peptide Ca^2+^-binding domain into the mNeonGreen fluorescent protein and performed several rounds of optimization using directed molecular evolution in a bacterial system [19]. We selected the mNeonGreen protein as the fluorescent moiety because of its high molecular brightness that was preserved in the NTnC calcium indicator [8]. As a Ca^2+^-binding motif, we used the CaM and M13-peptide excised from the GCaMP6s GECI and fused together via the flexible GGSGGGSSS linker in a similar way as in ncpGCaMP6s [17]. We started with the generation of two libraries with the insertion of the M13/CaM or CaM/M13 Ca^2+^-binding parts in a different order between residues 145 and 146 of the mNeonGreen protein in the same way as in the NTnC indicator (Figure 1) [8], and randomized both of the 3-amino-acid-long linkers between the fluorescent mNeonGreen and Ca^2+^-binding domains (Appendix A). These libraries were further analyzed using a two-step screening strategy, including screening on the bacterial colonies under a fluorescent stereomicroscope and in the solution in a 96-well plate format as described earlier [9]. As a result of screening, we found fluorescent variants with positive responses to calcium ions, demonstrating the maximal fluorescence contrasts of 1.3- and 2.1-fold for the M13/CaM- and CaM/M13-based libraries, respectively. The variant with the highest calcium sensitivity was chosen as a template for further optimization. 

The chosen clone was subjected to ten sequential rounds of random mutagenesis, followed by screening. During each round, we screened approximately 20,000 colonies to identify variants with the largest response to the calcium ions and the highest brightness, following the previously described protocol [9]. After ten rounds of random mutagenesis and selection, we chose the four variants with the best performance in terms of fluorescence contrast and brightness, but with different affinity to calcium ions, and named them NCaMP4, NCaMP7, NCaMP9, and NCaMP10 (mNeonGreen-derived CaM/M13-Peptide-based calcium indicator). The NCaMP indicators had 25–26 mutations relative to the original template-library (Appendix A).

### 2.2. Crystal Structure of the NCaMP7 Indicator in Its Calcium-Saturated State

The structural study of NCaMP7 was carried out to shed light on the molecular basis of the sensitivity of NCaMPs to calcium ions, to describe the impacts of mutations found during the engineering of the NCaMP indicators, and thus to help future research improve fluorescent sensors’ properties. We chose the NCaMP7 indicator for structural studies because it outperformed other NCaMPs in neuronal cultures and in in vivo experiments, as described below. We successfully crystallized the NCaMP7 indicator in the Ca^2+^-bound state and determined its spatial structure at 1.75 Å resolution.

There is one NCaMP7 molecule in the asymmetric unit, and contact analysis revealed that the protein has a monomeric state in the crystal. The overall structure of the NCaMP7 consists of two linked domains—the mNeonGreen fluorescent domain and the CaM/M13-peptide Ca^2+^-binding domain (Figure 1a and Figure 2a). Both domains are clearly visible in the electron density map. The mNeonGreen domain has a typical β-barrel fold with the chromophore formed by ^68^GYG^70^ amino acids and positioned on the central helix of the barrel. The chromophore orientation is additionally fixed by hydrogen bonds (H-bonds) to P65, R98, Y114, and E401, and water-mediated H-bonds to Y225, Y366, R386, and T388 (Figure 2b). The phenolic group of the chromophore points to the β-barrel surface towards the Y225, C330, and W348 residues. The Ca^2+^-binding domain is tightly bound to the fluorescent domain via a network of H-bonds and salt bridges. The M13-peptide of the Ca^2+^-binding domain is folded into α-helix and disposed of in the central part of this domain. Comparison of NCaMP7 (protein data bank identifier, PDB ID—6XW2) and GCaMP6m (PDB ID—3WLD) [20] structures being superposed by their calcium-binding parts (root-mean-square deviation, RMSD—1.6 Å^2^) revealed a different mutual orientation of both domains, despite a canonical overall architecture of the molecules (Appendix A). This difference may be attributed to the different NTnC- and GCaMP-like types of design for the indicators that implies different contact interfaces between the fluorescent domain and the CaM/M13-peptide calcium-binding domain. 

To understand the molecular bases of the Ca^2+^-dependent fluorescence changes for the NCaMP7 indicator, we first analyzed the direct contacts between the chromophore moiety and the Ca^2+^-binding domain. The deprotonated phenolic group of the chromophore forms three water-mediated H-bonds with the OH group of T388, the main chain C=O of R386, and the OH group of Y225 (Figure 2b). Notably, Y225 is located at the end of α4 in CaM and protrudes towards the surface “hole” of the NCaMP7 fluorescent part to stabilize the deprotonated phenolic group of the chromophore. Therefore, residues from both the fluorescent and Ca^2+^-binding domains stabilize the deprotonated state of the NCaMP7 chromophore. The deprotonated phenolic group of the chromophore in the parental mNeonGreen, in addition to two water-mediated H-bonds with R195 (corresponding to R386 in NCaMP7), forms also two water-mediated H-bonds with the main chain C=O of C139 and with the OH group of S141 (corresponding to C330 and S332 in NCaMP7) [21]. According to the GCaMP6m structure, two amino acid residues from α4 of CaM, R376, and Y380 (corresponding to the respective R221 and Y225 in NCaMP7; Appendix A), protrude towards the surface entrance of cpEGFP [20]. The former in GCaMP6m stabilizes the deprotonated phenolic group of the chromophore through a water-mediated H-bond; the corresponding R221 in NCaMP7 is located far away from the chromophore. The Y380 residue in GCaMP6m does not form an H-bond with the chromophore (like Y225 in NCaMP7), but provides a bulk phenolic group that blocks solvent access to the chromophore [20]. Hence, we speculate that Y225 in NCaMP7 (similar to Y380 in GCaMP6m [20]) plays a crucial role in the translation of the Ca^2+^-dependent conformational change in the CaM/M13-peptide domain to the chromophore’s environment, modulating indicator fluorescence.

We next describe NCaMP7 calcium coordinating centers, which determined the affinity and specificity to calcium ions, and compare their structure to that in the other Ca^2+^-binding proteins. Calcium ions in each of the four EF hands of the Ca^2+^-binding domain form one H-bond with a water molecule and six H-bonds with five amino acids (enumeration of amino acid positions starts from the first in the EF hand): D1 (one H-bond with carboxyl), D3 (one H-bond with carboxyl), D/N5 (one H-bond with carboxyl or amide), T/Y/Q7 (one H-bond with backbone C=O), and E12 (two H-bonds with carboxyl) (Figure 2c–e). Ca^2+^-bound water molecules in the EF hands form from two to four additional H-bonds with D3, D/N5, D9, and/or E12. Therefore, the EF hands in NCaMP7 have a canonical structure consisting of two short α-helices connected by a loop region, which, together with a single water molecule, provide seven ligands coordinating a single Ca^2+^ ion in a pentagonal bi-pyramidal coordination sphere. The calcium coordination centers in the other calcium indicators and the Ca^2+^-binding proteins have a similar composition and 3D structure [22,23]. Hence, NCaMP7 calcium coordinating centers share compositions and structures typical to the calcium-binding proteins, and the calcium-binding centers were not affected by insertion of the CaM/M13 domain into mNeonGreen.

Next, we related the impacts of the mutations found during mutagenesis of the NCaMPs to the X-ray data. The NCaMP indicators revealed 25–26 mutations relative to the original template library (Appendix A). Among these mutations, 11, 8-9, and 6 mutations were located in the fluorescent domain, the CaM/M13-peptide calcium-binding domain, and the linkers between these domains, respectively. According to the crystal structure of NCaMP7, the R331V and F346L mutations in the fluorescent domain were located about 6.0 Å from the GYG chromophore tripeptide, and all other mutations were more than 6.0 Å away from the chromophore. We speculate that the F346L mutation may be responsible for the slightly decreased brightness of the NCaMP4 indicator (Appendix A) because it reduced the hydrophobic pocket composed of side rings of F346 and W348 residues (Appendix A). Since the side group of the V331 residue is directed away from the chromophore, an effect of the R331V mutation on the chromophore’s properties is unlikely (Appendix A). Hence, except for the F346L mutation, all other mutations in the fluorescent domain that appeared during NCaMP’s development seem not to affect the fluorescent properties of the chromophore.

Mutations in CaM and the M13-peptide of NCaMPs may presumably affect the dynamic range and calcium affinities of the indicators. All NCaMPs share six identical mutations in the CaM domain and one identical mutation in the M13-peptide (Appendix A). The I210L and S248G mutations are located in the EF2 and EF3 Ca^2+^-binding hands, respectively (Appendix A). According to NCaMP7’s crystal structure, the main chain C=O and NH groups of L210 in EF2 form H-bonds with the main chain NH and C=O groups of I174 located in EF1 (Appendix A). The side chain of L210 is buried into a bulk hydrophobic pocket formed by the hydrophobic amino acids from EF1 and EF2 (F163, F166, I174, L179, M198, I199, L210, F215, M218, and M219) and from the M13-peptide (A317, I318, and L321) (Appendix A). Therefore, the I210L mutation might influence EF1–EF2 and CaM–M13-peptide interactions. According to NCaMP7’s crystal structure, the main chain NH group of G248 from EF3 forms an H-bond with the carboxyl of E251 (that forms two H-bonds with Ca^2+^ in EF3) and the C=O group of G248 forms H-bonds with main chain NH groups of E251 and L252 (Appendix A). The S248G mutation in NCaMPs results in a loss of the –CH_2_–OH group and a possible water-mediated H-bond with Q282 from EF4 or with water, which coordinates Ca^2+^ in EF3. Hence, the S248G mutation may impact the tuning of the EF3–EF4 interaction and contacts inside EF3. Of the rest of the four identical mutations in CaM of NCaMPs—Q196R, D197V, D269G, and M271L—the first three are located on the surface of CaM and lead to the addition of a positive charge (Q196R) and the removal of two negative charges (D197V and D269G). The main chain carbonyl group of L271 forms an H-bond with NH of the side chain indole of W308 from the M13-peptide (Appendix A). The side chain of L271 is buried into the hydrophobic pocket formed by L252, V255, M256, and L263 from CaM, as well as W308 and A311 from the M13-peptide. The identical T311A mutation in the M13-peptide for all NCaMPs results in the removal of the OH group from the same hydrophobic cluster. This mutation in the M13-peptide might adjust the affinity of NCaMP indicators to calcium ions in a similar way as for the FGCaMP indicator [24]. Thus, seven common mutations in all calcium indicators of the NCaMP series may be responsible for the tuning of the intramolecular interactions inside CaM or between CaM and the M13-peptide.

In addition to common mutations, each of the NCaMP indicators contain additional individual mutations in the CaM domain (Appendix A), which may be responsible for the observed differences in their calcium affinities. According to the K_d_ values, NCaMP4 has the highest affinity to Ca^2+^ (Table 1 and Appendix A). The N189S mutation in NCaMP4 is located before the Ca^2+^-binding EF2 hand. According to NCaMP7’s structure, N189 forms H-bonds with two water molecules on the surface of the indicator through its main chain and two H-bonds with R184, one water-mediated bond via side chain atom with main chain of R184 and another one with side chain of R184 (Appendix A). There is an H-bond network from R184 to T176 in the EF1 hand through G180, and water-mediated contact with E192 (all of these residues lay on one α-helix). Therefore, the N189S mutation in NCaMP4 may indirectly influence the position of amino acids in the EF1 hand and result in higher affinity to Ca^2+^ ions. NCaMP7 contains two mutations around the EF2 hand, N200I and T217A. According to NCaMP7’s structure, I200 is located on the surface of the indicator, and its main chain C=O forms two H-bonds with two water molecules and one water-mediated H-bond with the main chain C=O of D203 in the EF2 hand coordinating Ca^2+^ ion (Appendix A). The N200I mutation in NCaMP7 may result in a loss of an anticipated additional water-mediated H-bond between the side chains of N200 and D203. According to NCaMP7’s structure, the main chain NH group of A217 forms an H-bond with the main chain C=O of P213 in the EF2 hand, and the main chain C=O group of A217 forms water-mediated H-bonds with the main chain C=O group of A147 and the side chain NH of His148 imidazole in the first linker between the fluorescent and Ca^2+^-binding domains of NCaMP7 (Appendix A). Hence, the N200I and T217A mutations in NCaMP7 may affect the positions of amino acids in the EF2 hand and in the first linker between the fluorescent and Ca^2+^-binding domains. The side chains of amino acids in positions 156 and 256 mutated in NCaMP9 and NCaMP10, respectively (Appendix A), are part of the hydrophobic clusters. The mutation I156L in NCaMP9 may influence the location of the calcium-coordinating amino acids in the EF2 hand because, according to NCaMP7’s structure, the side chain of I156 makes hydrophobic interactions with F212, F215, and L216 from the EF2 hand or adjacent to it (Appendix A). Therefore, the mutation I156L in NCaMP9 may result in a decrease of its calcium affinity through the suggested mechanism. The impact from the M256L mutation in NCaMP10 may be similar to the impact from the M271L mutation discussed above, because the side chains of both amino acids are buried in the same hydrophobic pocket formed by the amino acids of CaM and the M13-peptide (Appendix A). The E214G mutation in NCaMP10 is the key mutation responsible for decreasing calcium affinity of the sensor due to a loss of two H-bonds between E214 and Ca^2+^ ion in the EF2 hand (Appendix A). Thus, the main impacts of additional mutations in the CaM domain of NCaMPs on their properties are ensured by indirect changes of the positions of calcium-coordinating amino acids in the EF1 and EF2 hands or by direct loss of H-bonds with Ca^2+^ in the EF2 hand.

### 2.3. In Vitro Characterization of the Purified NCaMP Indicators 

First, we characterized the spectral and biochemical properties of the purified NCaMP7 calcium indicator in the Ca^2+^-free (apo) and Ca^2+^-saturated (sat) states (Figure 1b-f and Table 1). In parallel, we characterized the properties of its derivatives, NCaMP4, NCaMP9, and NCaMP10, having higher or lower affinity to calcium ions (Appendix A). Later, when we carried out a side-by-side comparison of NCaMPs using stopped-flow fluorimetry, in cultured neurons and in vivo in mice, NCaMP7 showed superior performance; thus, we focused on NCaMP7.

At pH 7.2, NCaMPs_apo_ and NCaMPs_sat_ had absorption peaks at 400–403 and 505–509 nm, respectively (Figure 1b, Table 1, and Appendix A). The excitation maxima in apo- and sat-states were at 400-408 and 508-512 nm, respectively. When excited at 390 and 470 nm, NCaMPs_apo_ and NCaMPs_sat_ fluoresced with emission peaks at 520 and 520–522 nm, respectively (Figure 1c, Table 1, and Appendix A). Hence, the NCaMP variants exhibited almost identical spectral properties. The brightness of the NCaMPs_sat_ indicators in terms of the product of the extinction coefficient and quantum yield was 1.66–1.82-fold larger than that of the EGFP protein. In the absence of calcium ions, the brightness of NCaMPs_apo_ dropped as a result of a decrease of both quantum yield and extinction coefficient. The fluorescence ΔF/F dynamic range of the NCaMP indicators varied in the range of 32–99. The addition of 1 mM (physiological) concentration of Mg^2+^ ions decreased the ΔF/F dynamic range of NCaMPs down to 8.9–29 (Table 1 and Appendix A); however, in the latter conditions, the high ΔF/F dynamic range of the NCaMP7 indicator was preserved over 27 and was only 1.7-fold lower than that of GCaMPs or 27-fold larger compared to the NTnC indicator. 

As the intracellular pH varies from 5.0 in lysosomes to 7.5 in the cytosol [27], we studied the dependence of the fluorescence and dynamic range of the NCaMP indicators on pH. In the presence of 100 µM calcium ions, NCaMPs_sat_ exhibited a pH dependence of their fluorescence with p*K*_a_ values of 5.88–6.18, which were similar to the p*K_a_* values of GCaMP6s_sat_ (Figure 1d, Table 1, and Appendix A). The fluorescence of NCaMPs_apo_ had a bell-shaped pH dependence with p*K*_a1_ = 5.38–5.44 and p*K*_a2_ = 6.58–6.84. The different pH stability of NCaMPs in the Ca^2+^-bound and Ca^2+^-free states resulted in a dependence of its dynamic range on pH. The fluorescence and dynamic range of the control GCaMP6s calcium indicator with p*K*_a_ = 6.16 and 9.6 in sat- and apo-states, respectively, were also sensitive to pH variations. Thus, pH variations can contribute to the NCaMPs Ca^2+^ response. 

We further assessed the affinity of the NCaMP indicators to Ca^2+^ ions in the absence and presence of 1 mM Mg^2+^ (equivalent to 0.58–1 mM free Mg^2+^ ions concentration), a concentration that mimics that of 0.5–1–5 mM in the cytosol of mammalian cells [28,29]. According to the equilibrium binding titration experiments, NCaMPs demonstrated a K_d_ value of 66–204 nM (Figure 1e, Table 1, and Appendix A). These affinities match the Ca^2+^-free concentration changes from 50–100 nM up to 250–10,000 nM in the cytosol of mammalian cells [30,31]. The equilibrium Hill coefficients for the NCaMP indicators varied in the range of 1.7–2.2. They were slightly lower as compared to the Hill coefficient (*n* = 3.5) for the GCaMP6s indicator. The decreased cooperativity of Ca^2+^ binding by NCaMPs, as compared to GCaPM6s, could be explained by the different topology of these indicators. The addition of 1 mM Mg^2+^ ions increased their K_d_ values up to 82–306 nM or in 1.24–1.66-fold. The lowering of NCaMPs’ affinity in the presence of Mg^2+^ ions was similar to the 1.58-fold increase of the K_d_ value for the GCaMP6s indicator. The addition of 1 mM Mg^2+^ ions also lowered the Hill coefficient values by 1.22–1.36-fold, decreasing the cooperativity of NCaMPs’ interaction with calcium ions (Table 1 and Appendix A). Hence, the physiological concentrations of Mg^2+^ ions decreased the dynamic range and Ca^2+^ affinity of the NCaMP indicators, but in these conditions, NCaMP7 preserved a high dynamic range of 27 and a calcium affinity of 125 nM, which were 1.7- and 1.8-fold lower, respectively, relative to the same values for GCaMP6s indicator.

In size-exclusion chromatography, in the presence of Ca^2+^ ions, NCaMP7 eluted as a monomer at a concentration of 1.7 mg/mL (Appendix A), similar to NTnC and GCaMP6s [8]. NCaMP7 preserved its monomeric state, even at higher concentrations, because it crystallized as a monomer (Figure 1a). Monomeric proteins are less prone to cytotoxicity in mammalian cells [32] and allow labeling of individual proteins [33]. The latter possibility may ensure the direction of NCaMP7 to different places inside the cells, such as organelles, and spines and synapsis in neurons, by fusing NCaMP7 with various signal peptides and proteins.

Overall, the in vitro characterization revealed that NCaMPs had 1.66–1.82-fold improved brightness as compared to the EGFP protein, were sensitive to pH variations, and demonstrated a positive ΔF/F response to calcium ions of 8.9–29 with an affinity of 82–306 nM in the presence of physiological Mg^2+^ concentration. The main advantages of the NCaMP7 indicator demonstrating the best performance in vivo are a 1.7-fold higher brightness over the GCaMP6s indicator and a positive response and significant 27-fold enhanced dynamic range as compared to the inverted NTnC, the first mNeonGreen-based calcium indicator.

### 2.4. Characterization of the Kinetics for the NCaMP Calcium Indicators Using Stopped-Flow Fluorimetry

Since neuronal activity occurs with high rates, it was important to assess the Ca^2+^-association and -dissociation kinetics for the NCaMP indicators in tight comparison with fast GCaMP6s and GCaMP6f GECIs. The kinetic curves obtained for the control GECIs and NCaMP4 were mono-exponential (Appendix A). Other NCaMP indicators exhibited bi-exponential calcium dissociation kinetics, mono-exponential association kinetics at a low (300 nM) Ca^2+^ concentration, and bi-exponential association kinetics at Ca^2+^ concentrations of 700–1000 nM (Figure 1f, Appendix A, Table 1, Appendix A). The two exponents and the respective rate constants (k_1_ and k_2_) corresponded to rapid and slow binding processes. The rapid processes dominated and accounted for the major fluorescence changes in all cases. 

The Ca^2+^ dissociation half-times of 1.1 ± 0.1 and 1.3 ± 0.1 s for NCaMP7 and NCaMP4 were close to that for GCaMP6s (1.01 ± 0.04 s) (Figure 1f, Table 1, and Appendix A). Of all NCaMPs, NCaMP10 showed the fastest Ca^2+^ dissociation, with a half-time of 0.75 ± 0.04 s. This was 2.0-fold slower than the dissociation kinetics of GCaMP6f, which showed a half-time of 0.37 ± 0.04 s (Appendix A). Despite the relatively low affinity to calcium ions, NCaMP9 had the slowest dissociation kinetics (half-time of 1.6 ± 0.1 s), 1.6-fold slower as compared to GCaMP6s. Hence, according to the dissociation kinetics data, NCaMP4, NCaMP7, and NCaMP10 resembled GCaMP6s and GCaMP6m [34], while NCaMP9 was not optimal in this regard.

At calcium concentrations of 300–1000 nM, the association rates for NCaMP7 were similar to the respective rates for the control GCaMP6s GECI (Figure 1f and Appendix A). NCaMP4 demonstrated the highest association rates among all NCaMPs, 2.3–3.3-fold higher than those for GCaMP6s, which was in line with its largest affinity to calcium ions (Appendix A). As compared to GCaMP6s, NCaMP9 and NCaMP10 bound to calcium ions with 1.2–2.6-fold different rates depending on calcium concentration. Hence, according to the stopped-flow experiments, NCaMP indicators demonstrated fast dissociation–association kinetics, similar to that for the fast indicators from the GCaMPs family, and can be applied for further characterization in HeLa and neuronal cells. Calcium dissociation–association kinetics for the best NCaMP7 indicator were very close to the respective kinetics of GCaMP6s in spite of the 1.8-fold larger calcium affinity for NCaMP7.

### 2.5. Calcium-Dependent Response of the NCaMP Calcium Indicators in HeLa Mammalian Cells

To compare the performance of the green NCaMP indicators with the GCaMP6s GECI in mammalian cells, we co-expressed them with the red jRGECO1a GECI [35] in HeLa cells and characterized their localization, brightness, and response to the Ca^2+^ transients. We generated the NES-jRGECO1a-P2A-NCaMPs/GCaMP6s fusions with self-cleavable P2A peptide and transiently expressed them in HeLa cells (Figure 3a and Appendix A). The NCaMP variants revealed green fluorescence evenly distributed in the cells and co-localized with red fluorescence of the jRGECO1a indicator (Figure 3a). 

The addition of 2.5 µM ionomycin to the cell cultures resulted in an increase of NCaMPs’ green fluorescence (Figure 3a) with an average ΔF/F response of 2.0–22.5 (Appendix A). The ΔF/F response of 10.0 ± 2.4 for the NCaMP7 indicator was similar (*p* = 0.8015) to the response of 12.2 ± 3.1 for the control GCaMP6s. The ΔF/F responses of NCaMPs were equal (NCaMP4, *p* = 0.6032) or 3.2–7.3-fold larger (NCaMP7-10, *p* = 0.0079) as compared to the response of the jRGECO1a indicator (Appendix A). Except for NCaMP10, the ΔF/F responses for the NCaMP7, NCaMP9, and NCaMP4 GECIs in HeLa cells were 1.3–4.5-fold lower as compared to their dynamic ranges for the purified proteins (Table 1, Appendix A); the decrease of the NCaMPs’ dynamic range in HeLa cells correlated with their calcium affinity in vitro. We speculate that only NCaMP10 with a K_d_ of 306 nM was not bound to calcium ions at physiological concentrations of calcium ions in resting HeLa cells, and all other NCaMPs with lower K_d_ values of 82–173 nM were, to different extents, bound to the calcium ions. Hence, except for NCaMP4, all NCaMPs demonstrated high ΔF/F values of 10–23 in response to ionomycin-induced calcium concentration elevation in HeLa cells, which were similar (NCaMP7, *p* = 0.8015) or 1.6- (NCaMP10, *p* = 0.0159) and 1.8-fold (NCaMP9, *p* = 0.079) larger as compared to the GCaMP6s indicator.

To estimate the brightness of NCaMPs and GCaMP6s in HeLa cells, we normalized the maximally achievable green fluorescence of the NCaMPs and the GCaMP6s GECI to the maximal red fluorescence of jRGECO1a expressed in the same cells upon ionomycin administration (Appendix A). The brightnesses of NCaMP4 (*p* = 0.4127), NCaMP7 (*p* = 0.6667), and NCaMP9 (*p* = 0.8889) were similar to the brightness of GCaMP6s. Only the brightness of NCaMP10 was slightly higher (*p* = 0.0238), by 1.4-fold, than that for GCaMP6s. Hence, in HeLa cells, most NCaMPs had brightnesses similar to the brightness of GCaMP6s.

It was earlier suggested that the CaM/M13-peptide pair in the ncpGCaMP6s indicator may be less accessible for interaction with endogenous proteins as compared to the GCaMP6s GECI as a consequence of their different designs [17]. Because the NCaMP7 indicator has the same NTnC-like design as ncpGCaMP, we assessed the suggested reduction of interactions of the NCaMP7 indicator with the intracellular environment using fluorescence recovery after photobleaching (FRAP) of the NCaMP7 study in the cytosol of HeLa cells. FRAP experiments were performed on HeLa cells transiently transfected with NCaMP7 and the control GCaMP6s GECI. In a 60 s time scale, NCaMP7 and the GCaMP6s GECI showed the same (*p* = 0.9444) percent of immobile fractions at low physiological Ca^2+^ concentrations (NCaMP7, 8 ± 19%, vs. GCaMP6s, 3 ± 15%) (Fig 3b), which were not statistically different (*p* = 0.6825) from 0%. In a 60 s time scale, ionomycin-induced elevated Ca^2+^ concentrations resulted in similar (*p* = 0.2222) increases of immobile fraction until 18 ± 6% and 13 ± 4% for NCaMP7 and the GCaMP6s GECI, respectively; these percent values were statistically different (*p* = 0.0079) from 0%. In a 2.5 s time scale under low calcium concentration, NCaMP7 revealed a significantly 2.5-fold (*p* = 0.0079) less percent of immobile fraction of 16 ± 5% as compared to 40 ± 7% for GCaMP6s (Figure 3c, left). In the same time scale at elevated Ca^2+^ concentrations, NCaMP7 and GCaMP6 had similar (*p* = 0.9444) mobility (38 ± 5% vs. 35 ± 12% of immobile fractions for NCaMP7 and GCaMP6s, respectively) (Figure 3c, right). Hence, depending on the time scale and calcium concentrations, NCaMP7 demonstrated a similar or 2.5-fold higher mobility than the GCaMP6s GECI. These results support the idea that the insertion design or topology of the NCaMP7 indicator prevents its interactions with its intracellular surroundings to some extent.

As compared to GCaMP6s, the FGCaMP calcium indicator based on fungal Ca^2+^-binding proteins demonstrated higher mobility at physiological calcium concentrations in a 0.6 s time scale [24]. A novel design for the GCaMP-X calcium indicator was used to overcome the calcium channel perturbations induced by the calmodulin in GCaMP6m [36]. Thus, GCaMP-X design, the NTnC-like topology of the NCaMP7 indicator, and the application of CaM from fungus in the FGCaMP indicator are alternative strategies to reduce intracellular environment perturbations by GECIs.

### 2.6. Visualization of Spontaneous and Induced Neuronal Activity in Dissociated Culture Using NCaMP Indicators and Confocal Imaging

To estimate the ΔF/F responses of NCaMPs compared to GCaMP6s GECI, we used an external electric field for stimulation of dissociated neuronal cultures co-expressing green NCaMPs or the control GCaMP6s GECI, together with the red R-GECO1 indicator (Appendix A). The ΔF/F responses per 1 action potential (AP) for NCaMP4 (19 ± 13%) and NCaMP10 (13 ± 18%) were similar (*p* = 0.2494 and 0.0953, respectively) to the same characteristic for GCaMP6s (13 ± 8%). The ΔF/F responses per 1 AP for NCaMP7 (40 ± 9%) and NCaMP9 (36 ± 9%) were significantly 3-fold (*p* < 0.0001) larger than the respective response for GCaMP6s. Hence, on neuronal cultures, all NCaMPs robustly detected neuronal activity, although NCaMP7 was the best in this respect. Thus, we selected NCaMP7 for further validation in neuronal culture.

To validate the functionality of the best NCaMP7 indicator in neurons in more detail, we characterized its localization and fluorescence changes during spontaneous activity in primary mouse neuronal cultures. Accordingly, for localization imaging, we transiently transfected neuronal cultures with plasmids carrying the NCaMP7 green indicator together with the reference of near-infrared mIRFP [37] under the control of the CAG and Syn promoters, respectively. NCaMP7 demonstrated even distribution in neurons and their branches and was excluded from the nucleus (Figure 4a).

For functional imaging, we transduced neuronal cultures with rAAVs carrying CAG-NES-NCaMP7 and recorded the spontaneous activity of neurons between 12–19 days in vitro (DIVs). The maximal and averaged ΔF/F responses for the NCaMP7 indicator were about 3.8 and 3.4, respectively (Figure 4b,c). The averaged rise and decay half-times for NCaMP7 were 2.4 ± 0.7 and 5 ± 2 s, respectively. The kinetics and response of NCaMP7 during neuronal spontaneous activity were slightly different from those for GCaMP6s, which demonstrated average rise and decay half-times of 0.6 ± 0.2 and 2.5 ± 1.7 s [8], respectively, and an average ΔF/F of 1.28 ± 1.83 [9]. Overall, these data indicate that the NCaMP7 indicator was well localized in cultured neurons and reliably visualized the spontaneous and electrical field-evoked activity of neuronal cultures.

### 2.7. In Vivo Imaging of Neuronal Activity in the Hippocampus of Freely Moving Mice Using NCaMPs and an nVista Miniscope

To finally choose the best variant among the NCaMP indicators, we performed in vivo imaging with an nVista head-mounted miniscope, and compared kinetics, ΔF/F responses, and signal-to-noise ratios (SNRs) of NCaMPs during the visualization of spontaneous neuronal calcium activity in the CA1 field of the hippocampus of freely moving mice during the exploration of a circular track (Figure 5a). NCaMPs were delivered into the mouse hippocampus using rAAVs carrying NES-NCaMPs under the control of the CAG promoter. Based on the spike detection routine described earlier [9], we calculated characteristics averaged across all recorded neuronal activity during the exploration of the O-shaped track by mice. Except for NCaMP9, all NCaMPs demonstrated similar average rise and decay half-times of 0.8–0.9 and 2.0–2.3 s, respectively (Appendix A). NCaMP9 had the slowest kinetics with rise and decay half-times of 1.2 ± 0.8 and 2.5 ± 1.2 s, respectively. The longest decay half-time for NCaMP9 in vivo correlated with its slowest dissociation kinetics measured using stopped-flow fluorimetry in vitro (Appendix A). Among all NCaMPs, NCaMP7 revealed the best in vivo performance with respect to peak ΔF/F and SNR values (Appendix A). Overall, because, among all NCaMPs tested, NCaMP7 showed similar fast dynamics but higher ΔF/F and SNR values, we considered it as the indicator of choice for in vivo applications and further characterized it more carefully.

Then, we compared the performance of the NCaMP7 and GCaMP6s indicators during the recording of total (specific and nonspecific) neuronal activity in the CA1 field of the hippocampus using the same model and NVista miniscope as described above, but obtaining more data for better statistical comparison. We recorded the green fluorescence in the CA1 area of hippocampus with a head-mounted miniscope (Figure 5a) when mice were exploring the O-shaped track with landmarks (examples of recordings are shown in Figure 5b). Using MIN1PIPE procedure pipeline [38] and manual inspection, we successfully identified active cells and extracted respective calcium activity ΔF/F traces (Figure 5b). The ΔF/F response averaged across all recorded neuronal activity for NCaMP7 was 1.9 ± 1.6; it was similar (*p* = 0.0615) to the average ΔF/F response of 2.1 ± 1.8 for GCaMP6s (Figure 5c and Appendix A). The average rise and decay half-times for NCaMP7 (0.95 ± 0.52 and 3.0 ± 0.9 s, respectively) were 1.25- and 1.2-fold (*p* < 0.0001) longer as compared to the same half-times for GCaMP6s (0.76 ± 0.49 and 2.5 ± 0.9 s, respectively). Hence, according to in vivo NVista imaging data, NCaMP7 visualized the total (specific and nonspecific) neuronal activity in the hippocampus of mice with efficiency practically identical to GCaMP6s.

We next compared the ability of the NCaMP7 and GCaMP6s indicators to visualize the place-specific activity of CA1 cells using an nVista miniscope [9]. With this aim, using the NCaMP7 and GCaMP6s indicators, we correlated the neuronal calcium activity in the CA1 area of the hippocampus with the mouse movement in the O-shaped track with landmarks (Figure 5a). As a result, using both indicators, we identified the neurons that were specifically activated in certain parts of the track (one example of an NCaMP7-labeled place cell is shown in Figure 5d and Video S1). The ΔF/F responses of NCaMP7 (3.6 ± 1.5) averaged across space-specific activity of neuronal place cells were similar (*p* = 0.9338) to the respective responses for the GCaMP6s indicator (3.7 ± 0.9) (Figure 5e). Overall, among all NCaMPs, the NCaMP7 indicator demonstrated the best performance in visualization of total hippocampal neuronal activity, which was similar to the performance of GCaMP6s; NCaMP7 identified hippocampal place cells with a similar efficiency as the GCaMP6s indicator.

### 2.8. In Vivo Two-Photon Imaging of Neuronal Activity in the Visual Cortex of Awake Mice Using the NCaMP7 Indicator

To assess the best NCaMP7 indicator in standard two-photon in vivo application, we next carried out two-photon calcium imaging of the NCaMP7 indicator in the 2/3 layer (L2/3) of the primary visual cortex (V1) of awake head-fixed mice during the presentation of drifting grating as a visual stimulus. The NCaMP7 indicator was delivered into the brain of the P0 pups using an injection of rAAV particles. In the 5–7 weeks post-infection, we implanted cranial windows above the V1 brain area of the mice and identified cytosolic nuclei-excluded expression of the NCaMP7 indicator in neuronal bodies (15-25 µm in diameter) according to the green fluorescence in the L2/3 at up to 450 µm in depth (Figure 6a). A wavelength of 960 nm was optimal for two-photon excitation of the NCaMP7-expressing neurons. We further imaged the calcium neuronal activity during the presentation of a black PC monitor, and the same monitor with black–white gratings moving in eight different directions, as a visual stimulus for the NCaMP7-expressing mice. We found neurons demonstrating grating-evoked activity in three mice at depths of 100–230 µm (Figure 6b and Video S2).

According to the analysis among neurons with the largest grating-evoked responses, NCaMP7 showed a 3.4-fold higher (*p* < 0.0001) average ΔF/F response of 4.1 ± 1.1 compared to the ΔF/F value of 1.2 ± 0.6 for the GCaMP6s indicator (Figure 6c). As compared to GCaMP6s, the increased response of NCaMP7 during stimulus-evoked neuronal activity in the mouse cortex correlated with its 3-fold higher ΔF/F values during electrical stimulation of cultured neurons (Appendix A). NCaMP7 and GCaMP6s demonstrated similar SNR values of 16 ± 12 and 16 ± 14 (*p* = 0.8970), respectively (Figure 6c). Overall, these data show that the NCaMP7 indicator is appropriate for in vivo two-photon calcium imaging in the mouse cortex and enables the detection of stimulus-evoked calcium transients in neurons.

## 3. Materials and Methods 

### 3.1. Mutagenesis and Library Screening

Libraries construction using primers listed in Appendix A and screening were performed as described in reference [9].

### 3.2. Proteins Purification and Characterization

Proteins were expressed, purified, and characterized as described in reference [9]. 

To assess the equilibrium K_d_, the purified proteins (2 µg/mL) were added to buffer A (30 mM HEPES, 100 mM KCl, pH 7.2) supplemented with 10 mM EGTA (zero free Ca^2+^) pre-mixed in various ratios with buffer A supplemented with 10 mM Ca-EGTA (39 µM free Ca^2+^). After 20 min of equilibration at r.t. their green fluorescence was measured on ModulusTM II Microplate Reader (TurnerBiosystems, Sunnyvale, CA, USA). The titration of indicators with Ca^2+^ ions in the presence of 1 mM Mg^2+^ was done similarly, except buffer A supplemented with 1 mM MgCl_2_ and 10 mM EGTA (zero free Ca^2+^) was mixed with buffer A supplemented with 1 mM MgCl_2_, 10 mM Ca-EGTA (39 µM free Ca^2+^).

To determine the oligomeric state of the NCaMP7, the protein (1.7 mg/mL concentration) was applied to a Superdex 200 10/30 GL column (GE Healthcare, Chicago, IL, USA) equilibrated with 20 mM Tris-HCl, pH 7.5, 5 mM CaCl_2_.

### 3.3. Protein Purification for X-Ray Crystallography

The bacterial cells expressing the NCaMP7 protein with a His-tag and Tobacco Etch Virus (TEV) protease cleavage site were harvested by centrifugation, resuspended in 40 mM Tris-HCl buffer, pH 7.5 supplemented with 400 mM NaCl, 10 mM Imidazole, 0.2% Triton X-100, and 1 mM PMSF, and disrupted by sonication. The crude cell extract was centrifuged for 30 min at 28,000× *g* and 4 °C. The supernatant was applied to a Ni-NTA Superflow column (Qiagen, Hilden, Germany) equilibrated with the binding buffer (40 mM Tris-HCl, pH 7.5, containing 400 mM NaCl, 10 mM imidazole and 0.1% (*v*/*v*) Triton X-100). Washing and elution were performed with the same binding buffer without Triton X-100 and supplemented with 40 mM and 300 mM imidazole, respectively. Eluted protein was concentrated using a 10 kDa cutoff centrifugal filter device (Millipore, Burlington, MA, USA), and transferred into 40 mM Tris-HCl buffer, pH 7.5, containing 200 mM NaCl, 1 mM EDTA, 1 mM β-mercaptoethanol, 5 mM Imidazole and TEV protease (1 mg per 10 mg of protein). The solution was incubated overnight at +4 °C, dialyzed against the binding buffer, and applied to a Ni-NTA Superflow column (Qiagen, EU). TEV protease and cleaved His-tag were absorbed onto the column Ni-NTA Superflow column (Qiagen, EU) and the flow-through was concentrated, buffer exchanged to 20 mM Tris pH 7.5, 5 mM CaCl_2_ and applied to a ResourceQ column (GE Healthcare) equilibrated with the same buffer. Recombinant NCaMP7 was eluted with a linear gradient from 0 to 1M NaCl, concentrated and stored at −70 °C.

### 3.4. Crystallization of NCaMP7

An initial crystallization screening of NCaMP7 was performed with a robotic crystallization system (Rigaku, Tokyo, Japan) and commercially available 96-well crystallization screens (Hampton Research and Anatrace, Aliso Viejo, CA, USA) at 15 °C using the sitting drop vapor diffusion method. The protein concentration was 17 mg/ml in the following buffer: 20 mM Tris-HCl, 200 mM NaCl, pH 7.5 supplemented with 5 mM CaCl_2_. Optimization of the initial conditions was performed by the hanging-drop vapor-diffusion method in a 24-well VDX plates. Rod-like crystals were obtained within 3 weeks in the following conditions: 0.1 M Sodium acetate pH 4.6, 0.2 M ammonium sulfate, 22% (*w*/*v*) PEG 2000 MME. 

### 3.5. Data Collection, Processing, Structure Solution, and Refinement

The NCaMP7 crystals were briefly soaked in a 100% Paratone oil (Hampton research, Aliso Viejo, CA, USA) immediately prior to diffraction data collection and flash-frozen in liquid nitrogen. The X-ray data were collected from a single crystal at 100 K at the beamline K 4.4 of the Kurchatov SNC (Moscow, Russia). The data were indexed, integrated, and scaled using the XDS program, v. 15 March 2019 [39] (Appendix A). Based on the L-test [40] the dataset was not twinned. The Pointless program, v. 1.11.21 [41] suggested the P2_1_2_1_2_1_ space group.

The structure of NCaMP7 was solved by the molecular replacement method using the MOLREP program, v. 11.7.02 [42] and structure of the NTnC genetically-encoded green calcium indicator (PDB ID 5MWC) as an initial model. The refinement of the structure was carried out using the REFMAC5 program, v. 7.0.078, of the CCP4 suite [43]. Translation Libration Screw-motion refinement (TLS) was introduced at the earlier stages of refinement. The visual inspection of electron density maps and the manual rebuilding of the model were carried out using the COOT interactive graphics program, v. 0.8.9.2 [44]. The resolution was successively increased to 1.75 Å and the hydrogen atoms in the fixed positions were included during the final refinement cycles. In the final model, an asymmetric unit contained one independent copy of the protein of 404 residues with chromophore together with total 357 water molecules, 4 calcium ions, and 4 sulfate molecules from the crystallization solution. First, 11 residues from the *N*-terminal, as well as the last 10 residues from the *C*-terminal part of the protein were not visible in electron density, possibly due to their high flexibility.

### 3.6. Structure Analysis and Validation

The visual inspection of the structure was carried out using the COOT program and the PyMOL Molecular Graphics System, v. 1.9.0.0 (Schrödinger, LLC, New York, NY, USA). The structure comparison and superposition were made using the PDBeFold program, v. 2.59 [45], while contacts were analyzed using the PDBePISA, v. 1.52 [46] and WHATIF software [47].

### 3.7. Stopped-Flow Fluorimetry

The Ca^2+^-binding kinetics were analyzed using a Chirascan Spectrofluorimeter (Applied Photophysics, UK) equipped with a stopped-flow module as described previously [9]. 

### 3.8. Mammalian Plasmid Construction

In order to construct the pAAV-*CAG*-NES-NCaMPs plasmids, the NCaMPs genes were PCR-amplified as the BglII-EcoRI fragments using primers listed in the Appendix A, and swapped with the mCherry gene in the pAAV-*CAG*-NES-mCherry vector. In order to construct pAAV-*CAG*-NES-jRGECO1-P2A-NCaMPs and pAAV-*CAG*-NES-jRGECO1-P2A-GCaMP6s plasmids, NCaMPs and GCaMP6s genes were PCR amplified as the BglII-EcoRI fragments and swapped with the mCherry gene in the pAAV-*CAG*-NES-jRGECO1-P2A-mCherry vector.

### 3.9. Mammalian Live-Cell Imaging

HeLa Kyoto cell cultures were imaged 24–48 h after the transient lipofectamine transfection before and immediately after 2.5 µM Ionomycin addition using a laser spinning-disk Andor XDi Technology Revolution multi-point confocal system (Andor Technology, Belfast, UK) as previously described [9]. 

For FRAP experiments, HeLa Kyoto cells were treated as described above and imaged using a Leica SP5 STED confocal microscope (Leica-Microsystems, Bensheim, Germany) and 70% of 488 nm laser power for bleaching (power at 100%—65 mW) during 1000 ms, with the capture settings: 100 ms per frame, 20 and 600 frames before and after bleaching, respectively, resolution—16 × 16 pixels, pixel size—0.38 µm.

### 3.10. Imaging in Primary Mouse Neuronal Cultures

For dissociated hippocampal mouse neuron culture preparation, postnatal day 0 or 1 Swiss Webster mice (Taconic Biosciences, Rensselaer, NY, USA) were used as previously described [37]. Briefly, dissected hippocampal tissue was digested with 50 units of papain (Worthington Biochem, Lakewood, NJ, USA) for 6–8 min at 37 °C, and the digestion was stopped by incubation with ovomucoid trypsin inhibitor (Worthington Biochem, Lakewood, NJ, USA) for 4 min at 37 °C. Tissue was gently dissociated with Pasteur pipettes, and dissociated neurons were plated at a density of 25,000–30,000 per glass coverslip coated with Matrigel (BD Biosciences, Franklin Lakes, NJ, USA) in glass-bottom 24-well plates. Neurons were seeded in 100 µL of plating medium containing MEM (Life Technologies, Carlsbad, CA, USA), glucose (33mM; Sigma, St. Louis, MO, USA), transferrin (0.01%; Sigma), HEPES (10mM; Sigma), Glutagro (2mM; Corning, New York, NY, USA), insulin (0.13%; Millipore, Berlington, MA, USA), B27 supplement (2%; Thermo Fisher Scientific, Waltham, MA, USA), and heat-inactivated FBS (7.5%; Corning, New York, NY, USA). After cell adhesion, an additional 1 ml per well of plating medium was added. AraC (0.002mM; Sigma) was added when glia density was 50–70% of confluence, about 48 h after cell plating. Neurons were grown at 37 °C and 5% CO_2_ in a humidified atmosphere.

For confocal imaging, the neurons were transfected at 4–5 DIVs with a commercial calcium phosphate transfection kit (Life Technologies) as previously described [37]. Briefly, a mixture of the pAAV-CAG-NES-NCaMP7 and pAAV-Syn-miRFP plasmid DNA at 250 ng each per well was used for transfection followed by additional washing with acidic MEM buffer (pH 6.7–6.8) after 30–60 min of calcium phosphate precipitate incubation to remove residual precipitates. Neurons were imaged between DIV 11 and 19 DIV using an inverted Nikon Eclipse Ti microscope equipped with a spinning disk sCSUW1 confocal scanner unit (Yokogawa, Osaka, Japan), 488-, 561- and 642-nm solid state lasers, 525/25-nm and 664LP emission filters, a 40 × NA 1.15 water immersion objective (Nikon, Tokyo, Japan) and a 4.2 PLUS Zyla camera (Andor Technology, Belfast, UK), controlled by NIS-Elements AR software.

For functional imaging of calcium dynamics, cultured neurons were transduced at 4–5 DIV by administering ~1010 viral particles of rAAV/DJ-CAG-NES-NCaMP7 (Janelia Research Campus, Howard Hughes Medical Institute) per well (the rAAV genome titer was determined by dot blot). Neuron imaging was performed between DIV 12 and 19 DIV (~9–14 d post transfection) to allow for sodium channel maturation. Imaging was done using a Nikon Eclipse Ti inverted microscope equipped with a 40×NA 1.15 water immersion objective (Nikon), a SPECTRA X light engine (Lumencor), and OrcaFlash4.0v2 camera (Hamamatsu), controlled by NIS-Elements AR software.

Isolation, imaging, and electrical field stimulation of neuronal cultures was performed using a standard procedure, home-made pulse generator and Andor XDi Technology Revolution multi-point confocal system, respectively, as described previously [9].

### 3.11. Surgery and Imaging in Hippocampus Using an nVista HD Miniscope

Surgical procedures and imaging using the nVista HD miniature microscope (Inscopix Inc., Palo Alto, CA, USA) were described earlier [9]. Imaging data analysis was performed with custom MATLAB scripts, based on NoRMCorre [48] and MIN1PIPE [38] routines. First, imaging data was downsampled by factor 2 for increasing computation speed, and then displacements were corrected using NoRMCorre routine. After that, neuron location and traces were extracted using MIN1PIPE pipeline. Then, calcium events exceeding 4 MADs were detected using approaches described earlier [9]. Video tracking was performed via open-source visual programming media Bonsai [49]. Place fields were estimated using a conservative approach: The cell was considered spatial selective only if its activation was observed more than in 50% visits of putative place field zone.

### 3.12. Viral Injection to the Neonatal Mouse Brain and Surgery for in Vivo Two-Photon Imaging

P0 pups of C57BL/6 mice (Jackson Laboratory, Bar Harbor, ME, USA) were collected from the cage and prepared for viral injection by cryoanesthesia. The injection to neonatal brain was performed in accordance with the following protocol [50,51]. Briefly, rAAV were diluted in PBS containing 0.05% Trypan blue (NanoEnTek, Seoul, South Korea) and a volume of 2.5–7.5 µL of final solution was injected to the right hemisphere by 5 µL Hamilton syringe. The needle was held perpendicular to the skull surface to the depth of 1.5–2 mm. The pups then were placed on a warming pad for 3–5 min and transferred back to the mother cage. Surgery for in vivo two-photon imaging was performed 5–7 weeks post-infection as described in reference [9].

### 3.13. Two-Photon Imaging in V1

Surgeries and two-photon imaging using an Olympus MPE1000 two-photon microscope equipped with a Mai Tai DeepSee Ti:Sapphire femtosecond-pulse laser (Spectra-Physics, Santa Clara, CA, USA) and a water-immersion objective lens, 20× 1.05 NA (Olympus, Waltham, MA, USA) were performed as described earlier [9] with modifications concerning the image analysis. The images analysis for the extraction of ΔF/F neuronal activity traces was performed in MatLab R2019a (Academic License 40869378) using an open-source library, CaImAn [52]. The SNR values were calculated as (ΔF/F_peak_)/STD, where STD was the standard deviation averaged 10 s before and after the moving grating presentation.

### 3.14. Statistics

To estimate the significance of the difference between two values, we used the Mann–Whitney Rank Sum Test and provided *p*-values (throughout the text in the brackets) calculated for the two-tailed hypothesis. We considered difference as significant, if *p*-value was <0.05.

### 3.15. Ethical Approval and Animal Care

All methods for animal care and all experimental protocols were approved by the National Research Center “Kurchatov Institute” Committee on Animal Care (NG-1/109PR of 13 February 2020) and were in accordance with the Russian Federation Order Requirements N 267 M3 and the National Institutes of Health Guide for the Care and Use of Laboratory Animals. For experiments performed at Massachusetts Institute of Technology (MIT), all methods for animal care and use were approved by the MIT Committee on Animal Care and were in accordance with the National Institutes of Health Guide for the Care and Use of Laboratory Animals. Seventeen and twelve C57BL/6 mice were used in this study, ages ~2–4 months and P0–1 old, respectively. Mice were used without regard to gender.

## 4. Conclusions

In conclusion, we developed the brightest genetically encoded green NCaMP7 calcium indicator available based on the mNeonGreen fluorescent protein, characterized its properties in vitro, solved and analyzed its X-ray structure, and demonstrated its applicability for in vivo one- and two-photon calcium imaging. In vitro characterization of the NCaMP7 indicator showed that it had a 1.7-fold higher molecular brightness as compared to GCaMP6s. The analysis of NCaM7’s structure elucidated the crucial role of the Y225 residue located in the calcium-binding domain in signal transduction to the fluorescent domain. FRAP experiments revealed that the NTnC-like design of NCaMP7 indicator, to some extent, prevents interactions with the intracellular environment. The NTnC-like design of the NCaMP7 indicator may also be advantageous for targeting of the NCaMP7 indicator to different intracellular organelles [9]. The average specific ΔF/F response of NCaMP7 during space-evoked activity of neuronal place cells in the CA1 area of hippocampus was high (3.6 ± 1.5) and similar (*p* = 0.4334) to the average specific ΔF/F response of the NCaMP7 indicator (4.1 ± 1.1) during moving grating-evoked activity of neurons in the V1 visual cortex (Figure 5d and Figure 6c). Hence, the performance of the NCaMP7 indicator was appropriate for both one- and two-photon in vivo applications. In many cases, NCaMP7 demonstrated characteristics similar to those of the GCaMP6 GECI. We anticipate that further enhancement of the NCaMP7 calcium association–dissociation dynamics and sensitivity may be possible, owing to structure-guided mutagenesis, based on our X-ray data and GCaMP7 series mutations [53]. We anticipate that the NCaMP7 indicator will become a new standard calcium indicator in neuroscience labs around the world [54].

## Figures and Tables

**Figure 1 ijms-21-01644-f001:**
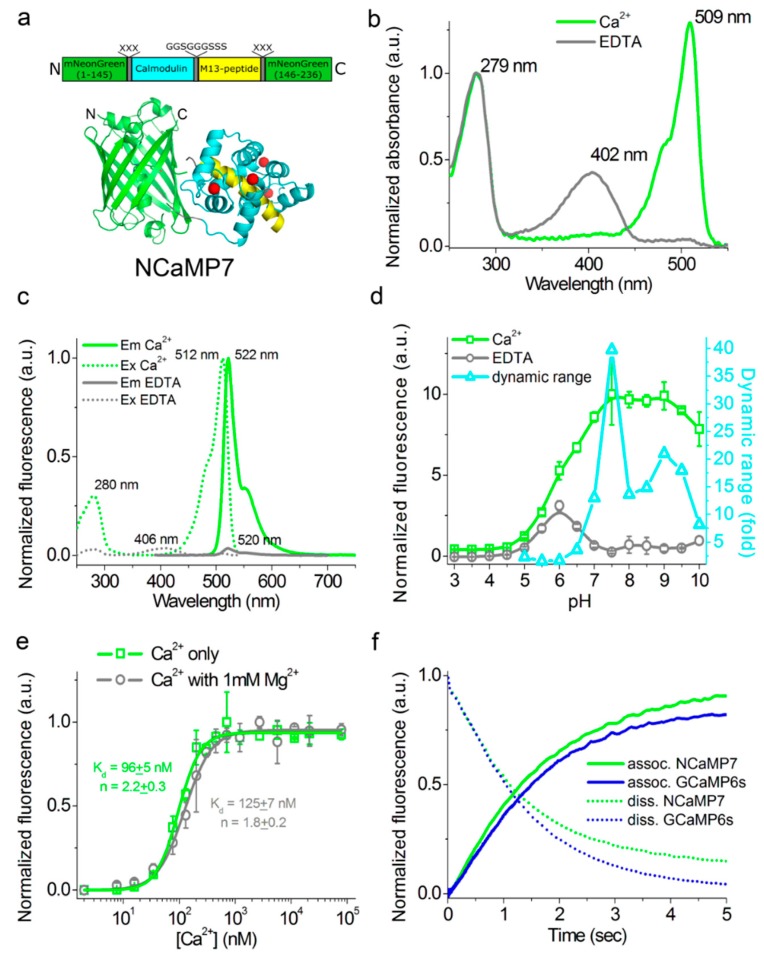
In vitro properties of the purified NCaMP7 indicator. (**a**) A scheme of the original library for optimization of linkers in the NCaMP7 indicator and a cartoon representation of its crystal structure (PDB ID—6XW2). (**b**) Absorbance spectra for NCaMP7 in Ca^2+^-bound (10 mM Ca^2+^) and Ca^2+^-free (10 mM EDTA) state at pH 7.2. (**c**) Excitation and emission spectra for NCaMP7 in Ca^2+^-bound (10 mM Ca^2+^) and Ca^2+^-free (10 mM EDTA) states, pH 7.2. (**d**) Fluorescence intensity for NCaMP7 in Ca^2+^-bound (10 µM Ca^2+^) and Ca^2+^-free (10 µM EDTA) states as a function of pH. Error bars represent the standard deviation. (**e**) Ca^2+^ titration curves for NCaMP7 in the absence and in the presence of 1 mM MgCl_2_, pH 7.2. Error bars represent the standard deviation. (**f**) Calcium-association and -dissociation kinetics for the NCaMP7 and control GCaMP6s indicators investigated using stopped-flow fluorimetry. Calcium-association and -dissociation kinetics curves were acquired at 300 nM final and at 1000 nM starting Ca^2+^-free concentrations, respectively. (**d**–**f**) Three replicates were averaged for analysis.

**Figure 2 ijms-21-01644-f002:**
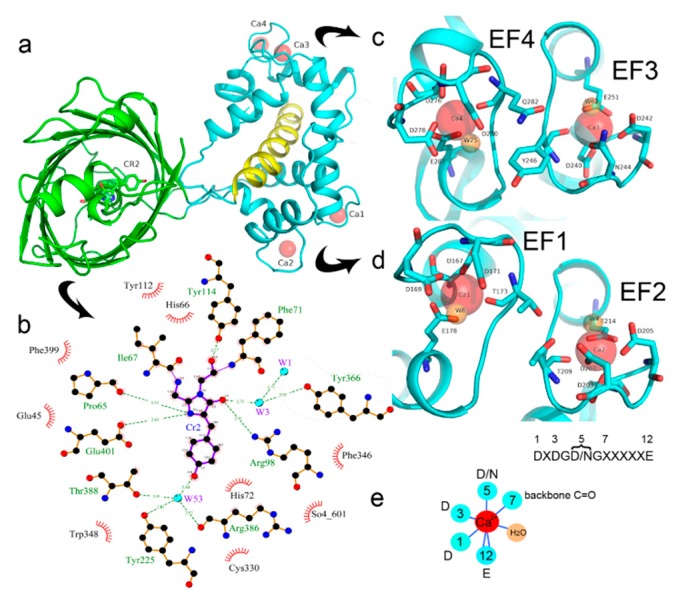
Overview of the crystal structure for the NCaMP7 indicator (PDB ID—6XW2). (**a**) Cartoon representation of NCaMP7 crystal structure (90 degrees rotated as compared to Figure 1a). The immediate surroundings of the chromophore (**b**) and calcium ions (**c**,**d**). (**e**) The coordination sphere of the calcium ion (in red) and positions for the calcium-coordinating residues in the sequence of EF1-4-hands loops (X means different residues observed in the same position among the four EF1-4-hands of NCaMP7). Hydrogen bonds and water molecule are shown as blue lines and orange circle, respectively.

**Figure 3 ijms-21-01644-f003:**
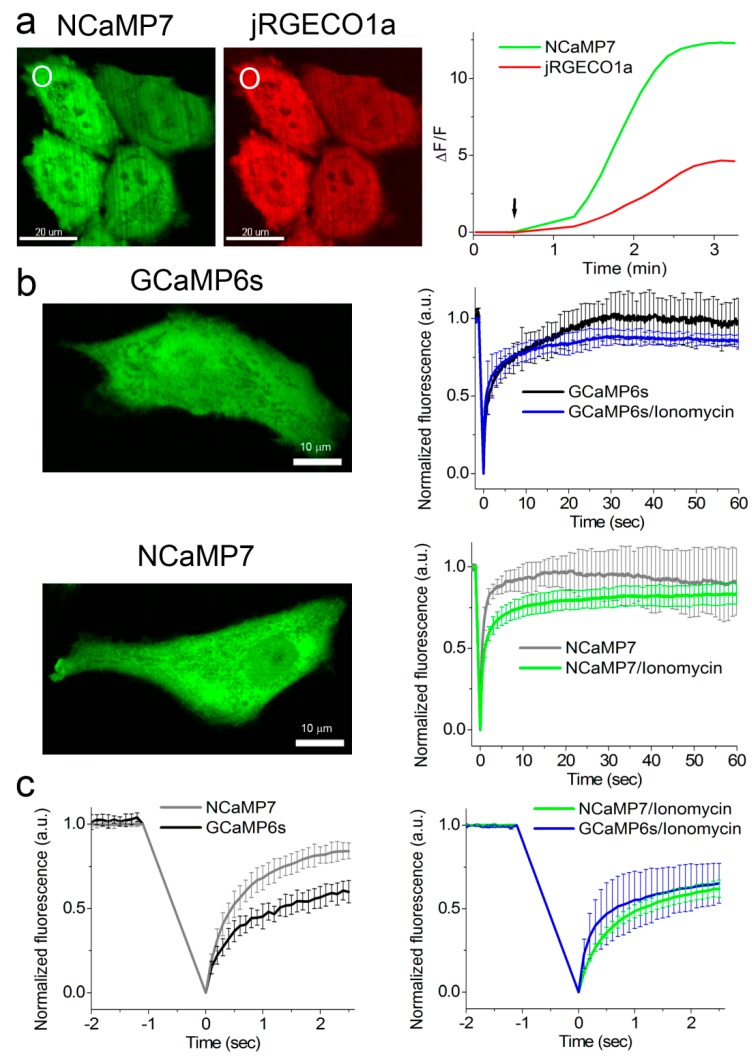
Response of the NCaMP7 indicator to Ca^2+^ variations in HeLa cells. (**a**) Confocal images of HeLa cells co-expressing green NCaMP7 (left) and red jRGECO1a (right) calcium indicators. The graph illustrates changes in green or red fluorescence of the NCaMP7 and reference co-expressed jR-GECO1a genetically encoded calcium indicators (GECIs) in response to the addition of 2.5 µM of ionomycin. The changes on the graph correspond to the area indicated with white circles. One example of five is shown. (**b**) Example of confocal images of HeLa cells expressing GCaMP6s and NCaMP7 calcium indicators used for the fluorescence recovery after photobleaching (FRAP) experiments. The graphs illustrate FRAP induced changes in green fluorescence of NCaMP7 and control GCaMP6s GECIs at physiological Ca^2+^ concentrations and in response to the 5 µM ionomycin addition for a 60 s time scale. (**c**) FRAP changes for 2.5 s time scale. (**b**,**c**) Error bars are standard deviations across five cells.

**Figure 4 ijms-21-01644-f004:**
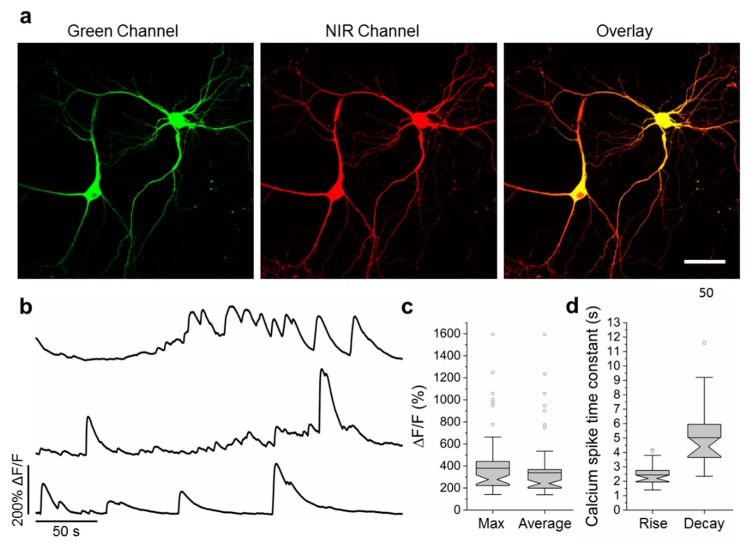
Calcium imaging of primary mouse neurons expressing NCaMP7. (**a**) Representative confocal images of neurons co-expressing NCaMP7 and miRFP. (**b**) Representative single cell recording of NCaMP7 green fluorescence responses during spontaneous neuronal activity. (**c**) Maximal (left) and average (right) ΔF/F for the experiment of **b**. (**d**) Time constant for the rise (left) and decay (right) of the NCaMP7 fluorescence during the calcium spikes for the experiment of **b**. Scale bar, 50 µm.

**Figure 5 ijms-21-01644-f005:**
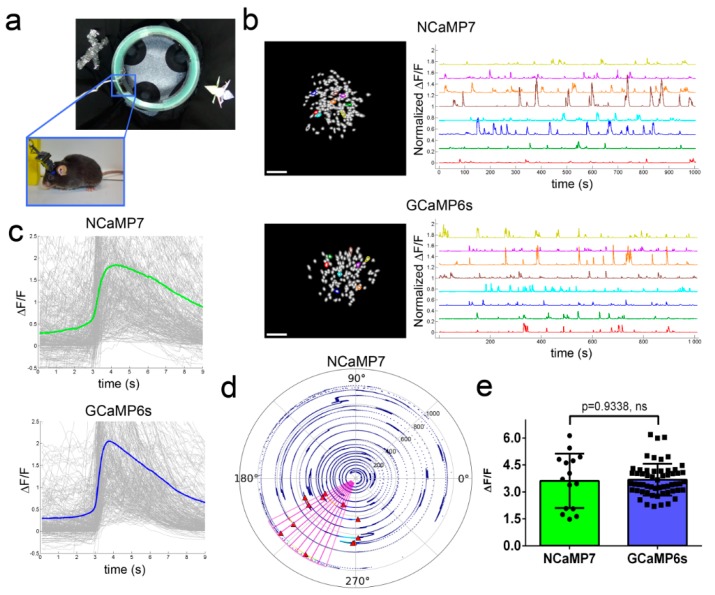
In vivo neuronal Ca^2+^ activity in the hippocampus of freely behaving mice visualized using NCaMP7 and GCaMP6s calcium indicators and a one-photon nVista HD miniscope. (**a**) Photo of O-shaped track with landmarks and mouse which explores it with an nVista HD miniscope mounted on its head. (**b**) Spatial filters and sample traces obtained from a 15-min imaging session of freely behaving mice expressing NCaMP7 and GCaMP6s GECIs. Scale bar, 100 µm. (**c**) Mean spikes for NCaMP7 and GCaMP6s calcium indicators; spikes above the 4 median absolute deviation (MAD) threshold, and not less than 50% of the maximal trace value, were aligned at the start of the peak (3 s). (**d**) Example of a circular plot for NCaMP7 mouse trajectory during the exploration of the circular track, synchronized with the spikes of a place cell (red triangles). (**e**) Averaged ΔF/F responses for space-evoked activity across place neuronal cells (*n* = 3, NCaMP7; *n* = 5, GCaMP6s) in the CA1 area of the hippocampus for the NCaMP7 and GCaMP6s indicators. The NCaMP7 and GCaMP6s indicators were delivered to the hippocampus with rAAVs carrying AAV-CAG-NES-NCaMP7/GCaMP6s. Ns, not significant.

**Figure 6 ijms-21-01644-f006:**
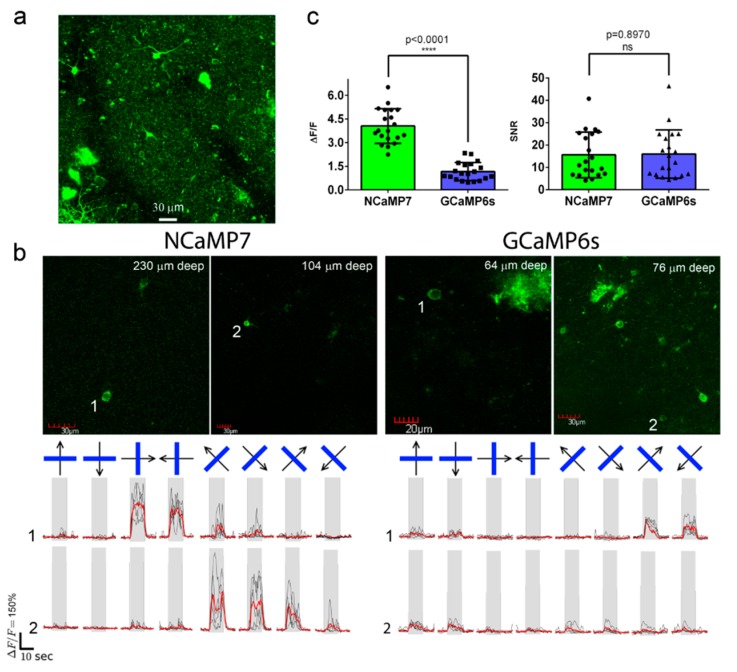
In vivo drifting grating-evoked neuronal activity in the mouse cortex visualized using NCaMP7, GCaMP6s calcium indicators, and two-photon microscopy. (**a**) The three-dimensional (3D) reconstruction of NCaMP7-positive cells in the V1 visual cortex area excited with 960 nm light. Block size, 360 × 360 × 470 µm. (**b**) Two-photon images of the V1 layer 2/3 neurons acquired during the presentation of drifting grating to the mice expressing NCaMP7 and GCaMP6s indicators. Raw (in black) and averaged (in red, averaged across five repetitions) ΔF/F responses during the presentation of drifting gratings (eight directions, five repetitions) are shown for the marked neurons. The directions of the drifting gratings (blue lines) are shown with arrows (in black). Grey vertical boxes correspond to the time of the grating presentation. (**c**) Averaged ΔF/F responses and SNR for grating-evoked activity across neurons (*n* = 2, NCaMP7; *n* = 2, GCaMP6s) in the V1 area for the NCaMP7 and GCaMP6s indicators.

**Table 1 ijms-21-01644-t001:** In vitro properties of purified NCaMP7 indicator compared to GCaMP6s.

Properties			Proteins				
			NCaMP7			GCaMP6s	
			apo		sat	apo	sat
Abs/Exc maximum (nm)			402/406		509/512	402/ND	500/ND
Emission maximum (nm)			520		522	518	515
Quantum yield ^a^			0.048 ± 0.003		0.52 ± 0.03	0.11 ± 0.01	0.61
ε (mM^−1^ cm^−1^) ^b^			46.6 ± 2.7		110.0 ± 7.3	33.3 ± 0.6	77 ± 3
Brightness (%) ^c^			6.5		179	8.3	107
ΔF/F (fold)		0 mM Mg^2+^	89 ± 27			43 ± 6	
		1 mM Mg^2+^	27 ± 3			46 ± 24	
p*K*_a_			5.43 ± 0.09 6.62 ± 0.09	6.18 ± 0.21		9.6 ± 0.3	6.16 ± 0.08
Kd (nM) ^d^	0 mM Mg^2+^		96 ± 5 (*n* = 2.2 ± 0.3)			144±3 (*n* = 3.5 ± 0.2)	
	1 mM Mg^2+^		125 ± 7 (*n* = 1.8 ± 0.2)			227.3 ± 0.2	
k_obs_ (s^−1^) ^e^			0.54 ± 0.02			0.49 ± 0.05	
k_off_ (s^−1^) ^f^	k_1_ (contrib., %)		0.89 ± 0.01 (78 ± 1)			0.69 ± 0.01	
	k_2_ (contrib., %)		0.11 ± 0.01 (22 ± 1)				
t_1/2_, s ^g^			1.1 ± 0.1			1.01 ± 0.04	

^a^ mEGFP (quantum yield, QY = 0.60 ref. [25]) and mTagBFP2 (QY = 0.64 ref. [26]) were used as reference standards for 500–509- and 402-nm absorbing states, respectively. ^b^ Extinction coefficient was determined by alkaline denaturation. ^c^ Brightness was normalized to mEGFP, with a QY of 0.60 and an extinction coefficient of 53.3 ± 3.6 mM^−1^cm^−1^. ^d^ Hill coefficient is shown in brackets. ^e^ The observed association rates were determined at 300 nM Ca^2+^ concentration from association kinetics curves (Figure 1f). GCaMP6f had k_obs_ value of 1.28 ± 0.03 sec^−1^. ^f^ k_off_ values were estimated from calcium dissociation curves (Figure 1f) using mono or double exponential decay fitting with individual exponent contributions shown in the brackets. GCaMP6f had a k_off_ value of 1.89 ± 0.01 s^−1^. ^g^ GCaMP6f had a t_off_ value of 0.37 ± 0.04 s. ND, not determined.

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
