# Peer review of "Novel Genetically Encoded Bright Positive Calcium Indicator NCaMP7 Based on the mNeonGreen Fluorescent Protein"

_ijms, 2020, doi:10.3390/ijms21051644_

Round 1

Reviewer 1 Report

This paper describes the development, characterization and testing of a new genetically encoded calcium indicator.   It is a complete study and well-organized paper describing first the rationale for the development, then the in vitro characterization, including the crystal structure, and finally in vivo testing.  This sensor looks very promising and will be of great interest to the field.

The paper needs some English editing, there are missing or extra articles, subject-verb mismatches, and other grammar errors.  Some examples:

  • In Table S7, the first three properties should not have the article A (it should say “number of mice” not “A number of mice”).
  • In several places it says “in the extent” which should be “to the extent”
  • Line 226 says: Hence, except F346L mutation, all other mutations in the fluorescent domain appeared during NCaMPs development seems not to affect fluorescent properties of the chromophore. It should say:  Hence, except for the F346L mutation, all other mutations in the fluorescent domain found during NCaMPs development seem not to affect the fluorescent properties…
  • Line 264: Except common mutations…. Do you mean: In addition to common mutations?

There are many more examples.

Author Response

Response to Reviewer 1 Comments

We thank reviewer 1 for his valuable comments and useful suggestions, which we have addressed entirely in the revised manuscript.

Reviewer #1:

This paper describes the development, characterization and testing of a new genetically encoded calcium indicator.   It is a complete study and well-organized paper describing first the rationale for the development, then the in vitro characterization, including the crystal structure, and finally in vivo testing.  This sensor looks very promising and will be of great interest to the field.

The paper needs some English editing, there are missing or extra articles, subject-verb mismatches, and other grammar errors.  Some examples:

Point 1:       In Table S7, the first three properties should not have the article A (it should say “number of mice” not “A number of mice”).

Response 1: In the revised manuscript, Table S7, we deleted the article A.

Point 2:       In several places it says “in the extent” which should be “to the extent”

Response 2: In the revised manuscript, we replaced “in the extent” with “to the extent”.

Point 3:       Line 226 says: Hence, except F346L mutation, all other mutations in the fluorescent domain appeared during NCaMPs development seems not to affect fluorescent properties of the chromophore. It should say:  Hence, except for the F346L mutation, all other mutations in the fluorescent domain found during NCaMPs development seem not to affect the fluorescent properties…

Response 3: In the revised manuscript, line 249, we replaced “Hence, except F346L mutation, all other mutations in the fluorescent domain appeared during NCaMPs development seems not to affect fluorescent properties of the chromophore.” with “Hence, except for the F346L mutation, all other mutations in the fluorescent domain that appeared during NCaMPs development seem not to affect the fluorescent properties of the chromophore.”.

Point 4:       Line 264: Except common mutations…. Do you mean: In addition to common mutations?

Response 4: In the revised manuscript, line 290, we changed “Except common mutations…  “ to “In addition to common mutations…”.

Point 5:       There are many more examples.

Response 5: To address this point we sent the manuscript to the MDPI English editing service.

Reviewer 2 Report

In this study, Subach et al. developed novel genetically encoded calcium indicators with a bright fluorescent protein mNeonGreen by making use of calmodulin-M13 as a Ca2+ sensing module. The authors solved the crystal structure of the sensor and inspected the molecular basis of the sensor. They also applied the sensor for neuronal cell imaging in primary culture as well as in vivo. Although the impact of the new sensor was not emphasized in the manuscript, the reviewer acknowledges that this report completed a series of study from sensor designing and investigation of the structural basis to the application in vivo, which is worth publishing in the International Journal of Molecular Science. The reviewer still find following points to be addressed by authors prior to considering the publication.

  1. This manuscript has no “discussion” section. Which is a bit strange to the reviewer. Crucial and sufficient discussion should be provided based on results.
  2. The authors are emphasizing the positive response of NCaMP7 with ΔF/F of 27 as the advantage of this sensor over NTnC, an older version of GECI with mNeonGreen, which shows inverted response with ΔF/F of 1. The positive response of the new sensor is probably an advantage. However, this reviewer feels the value ΔF/F inappropriate to compare sensors that show positive and negative response. In the case of positive sensor, denominator (namely F) becomes close to zero, which makes ΔF/F value large even if the ΔF is small. On the other hand, in the case of negative sensor, maximum response makes signal close to zero and hence ΔF ≈ F results in ΔF/F=1. Hence, the meaning of ΔF/F=1 is essentially different between positive and negative sensor. Keep in mind that ΔF itself is essential for detection of response. The reviewer believe that ΔF/F should not be emphasized to compare these two sensors.
  3. The advantage of NCaMP7 over GCaMP6s is not very well emphasized. The performance of these two sensors looked similar, and the reviewer recognized only one advantage, which is in the case of grating-evoked neuronal activity. For a paper reporting novel sensors, advantage(s) of them should be clearly emphasized, ideally together with drawbacks of them.
  4. The authors are giving Ca2+ titration curves in the presence and absence of Mg2+. First of all, detailed information of Ca2+ buffer system is missing. Secondly, Mg2+ can influence upon the buffering system even if the chelator chosen has lower affinity against Mg2+ such as EGTA. For example, in the case of EGTA, magnesium stability constant is 5.21. Since Ca2+ buffering system relies on the equilibrium Ca2+ and EGTA binding, free EGTA concentration is crucial to determine free Ca2+ concentration. The concentration of free EGTA is usually a few mM. However, if 1 mM of Mg2+ is added to the system, Mg2+ binds to the free EGTA and significantly reduces the concentration of this species, which has a strong impact to the free Ca2+ concentration. The authors are describing difference in Kd values of the sensor, but it can include a big error if the impact of Mg2+ to the Ca2+ buffering system is not properly corrected. Furthermore, since large amount of Mg2+ is chelated to EGTA, free Mg2+ concentration in the system should be much lower than 1 mM, hence it is not possible to represent physiological free Mg2+ concentration.
  5. The authors performed size exclusion chromatography to evaluate a possible dimer formation of the sensor, and a calibration curve is shown in Figure S14. However, this calibration curve looked strange. Here the graph for calibration is drawn as molecular weight (linear scale) vs elution volume, and a bending line is provided on it. With this bending curve, it is not possible to determine molecular weight from the elution volume. In general, calibration curve of size exclusion chromatography is given by plotting logarithmic of molecular weight vs elution volume (or relative elution volume), so that linear regression is expected.
  6. In line 346, there is a following statement: “Monomeric proteins are …… and allow labeling of individual proteins [31]”. This is probably correct as a general statement, but it is not clear to the reviewer why the author is mentioning about this point for the calcium indicator. Are the authors aiming to use the sensor for the “labeling of individual proteins”?

Author Response

Response to Reviewer 2 Comments

We thank reviewer 2 for his valuable comments and useful suggestions, which we have addressed entirely in the revised manuscript.

Reviewer #2:

In this study, Subach et al. developed novel genetically encoded calcium indicators with a bright fluorescent protein mNeonGreen by making use of calmodulin-M13 as a Ca2+ sensing module. The authors solved the crystal structure of the sensor and inspected the molecular basis of the sensor. They also applied the sensor for neuronal cell imaging in primary culture as well as in vivo. Although the impact of the new sensor was not emphasized in the manuscript, the reviewer acknowledges that this report completed a series of study from sensor designing and investigation of the structural basis to the application in vivo, which is worth publishing in the International Journal of Molecular Science. The reviewer still find following points to be addressed by authors prior to considering the publication.

Point 1:       This manuscript has no “discussion” section. Which is a bit strange to the reviewer. Crucial and sufficient discussion should be provided based on results.

Response 1: Since the results were tightly combined with discussion in the fist version of submitted manuscript, in the revised manuscript we renamed “2. Results” section to “2. Results and Discussion” one and we want to keep results and their discussion in one section to avoid dramatic changes in the original manuscript.

Point 2:       The authors are emphasizing the positive response of NCaMP7 with ΔF/F of 27 as the advantage of this sensor over NTnC, an older version of GECI with mNeonGreen, which shows inverted response with ΔF/F of 1. The positive response of the new sensor is probably an advantage. However, this reviewer feels the value ΔF/F inappropriate to compare sensors that show positive and negative response. In the case of positive sensor, denominator (namely F) becomes close to zero, which makes ΔF/F value large even if the ΔF is small. On the other hand, in the case of negative sensor, maximum response makes signal close to zero and hence ΔF ≈ F results in ΔF/F=1. Hence, the meaning of ΔF/F=1 is essentially different between positive and negative sensor. Keep in mind that ΔF itself is essential for detection of response. The reviewer believe that ΔF/F should not be emphasized to compare these two sensors.

Response 2: For the comparison of dynamic ranges of NCaMP7 and NTnC indicators, we calculated DF/F values using the same equation: DF/F=(Fmax-Fmin)/Fmin, where Fmin and Fmax are minimal and maximal fluorescence values achievable in response to calcium free concentrations changes from zero to 39 µM. This equation can be applied for both positive and inverted indicators, since the DF value was normalized to the minimally achievable fluorescence for both types of indicators in the same manner. We are confident that DF/F values calculated as described here can be used for the comparison of these two sensors.

Point 3:       The advantage of NCaMP7 over GCaMP6s is not very well emphasized. The performance of these two sensors looked similar, and the reviewer recognized only one advantage, which is in the case of grating-evoked neuronal activity. For a paper reporting novel sensors, advantage(s) of them should be clearly emphasized, ideally together with drawbacks of them.

Response 3: In the “4. Conclusions” section we have made every effort to emphasize the advantages of NCaMP7 over GCaMP6s in a condensed form. Honestly, we could not find any drawbacks of NCaMP7 vs GCaMP6s.

Point 4.1:       The authors are giving Ca2+ titration curves in the presence and absence of Mg2+. First of all, detailed information of Ca2+ buffer system is missing. 

Response 4.1: In the revised manuscript, “3.2 Materials and Methods” section we added detailed information about Ca2+ buffer system: “To assess the equilibrium Kd, the purified proteins (2µg/ml) were added to buffer A (30 mM HEPES, 100 mM KCl, pH 7.2) supplemented with 10 mM EGTA (zero free Ca2+) pre-mixed in various ratios with buffer A supplemented with 10 mM Ca-EGTA (39 µM free Ca2+). After 20 min of equilibration at r.t. their green fluorescence was measured on ModulusTM II Microplate Reader (TurnerBiosystems, USA). Titration of indicators with Ca2+ ions in the presence of 1 mM Mg2+ was done similarly except buffer A supplemented with 1 mM MgCl2 and 10 mM EGTA (zero free Ca2+) was mixed with buffer A supplemented with 1 mM MgCl2, 10 mM Ca-EGTA (39 µM free Ca2+).”

Point 4.2:       Secondly, Mg2+ can influence upon the buffering system even if the chelator chosen has lower affinity against Mg2+ such as EGTA. For example, in the case of EGTA, magnesium stability constant is 5.21. Since Ca2+ buffering system relies on the equilibrium Ca2+ and EGTA binding, free EGTA concentration is crucial to determine free Ca2+ concentration. The concentration of free EGTA is usually a few mM. However, if 1 mM of Mg2+ is added to the system, Mg2+ binds to the free EGTA and significantly reduces the concentration of this species, which has a strong impact to the free Ca2+ concentration. The authors are describing difference in Kd values of the sensor, but it can include a big error if the impact of Mg2+ to the Ca2+ buffering system is not properly corrected.

Response 4.2: We estimated the impact of 1 mM Mg2+ ions to the concentration of free calcium ions in our system (10 mM EGTA/10mM CaEGTA) using online “Ca/Mg/ATP/EGTA Calculator v2.2b” calculator available via a link: https://somapp.ucdmc.ucdavis.edu/pharmacology/bers/maxchelator/CaMgATPEGTA-NIST-Plot.htm as well as according to the paper Tsien R. et al. Methods in enzymology, 1989. In the range of 0.25-10 mM CaEGTA concentrations used to estimate Kd values for the sensors, addition of 1 mM Mg2+ ions had practically no effect on the free calcium ions concentration (it increased only slightly by 3.7-4.5%). Thus, we concluded that we can neglect the impact of the 1 mM Mg2+ ions addition on the free calcium ions concentration.

Point 4.3:   Furthermore, since large amount of Mg2+ is chelated to EGTA, free Mg2+concentration in the system should be much lower than 1 mM, hence it is not possible to represent physiological free Mg2+ concentration.

Response 4.3: We estimated the variation of the free Mg2+ ions concentration in our system (10 mM EGTA/10mM CaEGTA) using online “Ca/Mg/ATP/EGTA Calculator v2.2b” calculator available via a link: https://somapp.ucdmc.ucdavis.edu/pharmacology/bers/maxchelator/CaMgATPEGTA-NIST-Plot.htm as well as according to the paper Tsien R. et al. Methods in enzymology, 1989. Accordingly, the free Mg2+ ions concentration varied from 0.58 mM (at 10 mM EGTA) till 0.9999 mM (at 10 mM CaEGTA). Hence, the free Mg2+ ions concentration varied in our system but in the range of physiological free Mg2+ ions concentration.

Accordingly, in the revised manuscript, in the main text, lines 300-302, we added the text highlighted with blue: “We further assessed the affinity of the NCaMPs indicators to Ca2+ ions in the absence and in the presence of 1 mM Mg2+ (equivalent to 0.58-1 mM free Mg2+ ions concentration), a concentration that mimics that of 0.5 - 1-5 mM in the cytosol of mammalian cells.”

Point 5:       The authors performed size exclusion chromatography to evaluate a possible dimer formation of the sensor, and a calibration curve is shown in Figure S14. However, this calibration curve looked strange. Here the graph for calibration is drawn as molecular weight (linear scale) vs elution volume, and a bending line is provided on it. With this bending curve, it is not possible to determine molecular weight from the elution volume. In general, calibration curve of size exclusion chromatography is given by plotting logarithmic of molecular weight vs elution volume (or relative elution volume), so that linear regression is expected.

Response 5: In the revised manuscript, Supplementary Information, we modified Figure S14 according to the reviewer’s comment and added text to the figure legend highlighted with blue color: “The molecular weight of NCaMP7 was determined from a linear regression of the dependence of logarithm of control molecular weights vs elution volume.” Note, that molecular weight of NCaMP7 of 52 kDa determined from linear regression was very close to the molecular weight of NCaMP7 of 54 kDa calculated without any assumptions of linearity.

Point 6:       In line 346, there is a following statement: “Monomeric proteins are …… and allow labeling of individual proteins [31]”. This is probably correct as a general statement, but it is not clear to the reviewer why the author is mentioning about this point for the calcium indicator. Are the authors aiming to use the sensor for the “labeling of individual proteins”?

Response 6: In the revised manuscript, main text, lines 412-414 we added the text highlighted with blue: “The latter possibility may ensure the direction of NCaMP7 to different places inside the cells, such as organelles, spines and synapsis in neurons, etc., by fusing NCaMP7 with various signal peptides and proteins.”